# Private and Non-private Uniformity Testing for Ranking Data

**Róbert Busa-Fekete**
Google Research, New York, USA
`busarobi@google.com`

**Dimitris Fotakis**
National Technical University of Athens, Greece
`fotakis@cs.ntua.gr`

**Manolis Zampetakis**
University of California, Berkeley, USA
`mzampet@berkeley.edu`

## Abstract

We study the problem of uniformity testing for statistical data that consists of rankings over $m$ items, where the alternative class is restricted to Mallows models. Testing ranking data is challenging because of the size of the large domain that is factorial in $m$, therefore the tester needs to take advantage of some structure of the alternative class. We show that uniform distribution can be distinguished from Mallows model with $O(m^{-1/2})$ samples based on simple pairwise statistics, which allows us to test uniformity using only two samples, if $m$ is large enough. We also consider uniformity testing with central and local differential privacy (DP) constraints. We present a central DP algorithm that requires $O(\max\{1/\epsilon_0, 1/\sqrt{m}\})$, where $\epsilon_0$ is the privacy budget parameter. Interestingly, our uniformity testing algorithm is straightforward to apply to the local DP scenario, since it works with binary statistics that is extracted from the ranking data. We carry out large-scale experiments, including $m = 10,000$, to show that our uniformity testing algorithms scale gracefully with $m$.

## 1 Introduction

Testing whether the data conforms with a model is a fundamental problem in data analysis with large number of applications in machine learning and data science. A special case of testing is the uniformity testing, i.e. to distinguish between the case that an unknown distribution accessible via samples is uniform versus $\epsilon$-far from uniform. Uniformity testing of discrete distribution has a long history with several applications, and it is well-understood [3, 7, 11, 12, 29, 31] as well as under differential privacy constraints [1, 2, 4].

In this paper, we assume that the statistical data consists of rankings over $m$ items. The testing problem that we study is to decide whether the data is generated from the uniform distribution over rankings, i.e. the probability of observing any ranking is $1/m!$, or from a distribution that is $\epsilon$-far from the uniform one in terms of total variation distance. In general, this testing problem cannot be tackled based on polynomial sample complexity, because the domain size for ranking data is $m!$. Therefore, we restrict the alternative hypothesis class to the Mallows model, introduced by [25], a.k.a. the exponential family on rankings. The Mallows model is widely used in ranking statistics and machine learning. The model has two parameters, the *central ranking* $\pi_0 \in \mathbb{S}_m$ and the *spread parameter* $\phi \in [0, 1]$. Based on these, the probability of observing a ranking $\pi \in \mathbb{S}_m$ is proportional to $\phi^{d(\pi, \pi_0)}$, where $d$ is a ranking distance, such as the number of discordant pairs, a.k.a. Kendall's tau distance. There are many applications of the Mallows model in Machine Learning, to name a few, label ranking [24], online learning [9], recommendation systems [30, 23] and clustering [27].

35th Conference on Neural Information Processing Systems (NeurIPS 2021).

Testing is a central problem in analysing output of ranking systems where the goal is to decide whether the output ranking data deviates from some expected behaviour, or is biased towards some group of object to be ranked or it is indeed fair ([28, 25, 32] and see Chapter 3-4 of [26]). Significant attention has been attracted to machine learning systems and their ability to discriminate against minorities, historically disadvantaged populations, and other protected groups when allocating resources (e.g., loans) or opportunities (e.g., jobs). For example, the Fair Lending Act – a commonly used fairness requirement introduced in the USA – is explicitly created to impose such unbiasedness on financial companies. This policy even has to be respected in audience targeting in online advertising. Uniformity and parameter testing can be applied to practical settings where we want to collect/provide evidence that our ranking system has no bias towards some group of users.

In this work, we propose two tests to tackle uniformity testing with the Mallows model as alternative class of ranking distributions: one of these tests works with just two samples when the number of items $m$ is large enough, and a more general one, with sample complexity $O(1/\sqrt{m})$, works for arbitrary $m$.

There are several applications where the data to be analyzed contains sensitive information. However, the data owner is willing to release some results of analysis based on the data, without revealing information of the individuals. It may thus be of importance to guarantee that working with sensitive data needed to test a statistical hypothesis protects sensitive information about the individual records in the dataset. Differential privacy (DP) is one of the most commonly used privacy preserving framework [15, 17] which has been adopted by several companies including Google [18], and Apple [13]. Differential privacy requires that the output of an algorithm has to be statistically close on two similar datasets $D$ and $D'$ that differ in the value of one element which a ranking in our case. We consider the two most common version of DP: central and local DP. In the central model, a trusted curator stores the database, and she runs the algorithm which has DP guaranty to analyze the sensitive data and then the output of the analysis, i.e. accept or reject in case of hypothesis testing, is released to the public [16]. Several alternatives to the central model have been proposed that relax the requirement that the users trust the curator to store their private data, for example local DP (LDP). In the local model, each user adds noise to her own data and responds to the analyst directly [19]. Interestingly, our uniformity testing algorithms can be easily applied in the LDP setting. We can summarize our main results as follows:

- We devise a uniformity test which works based on two samples when $m$ is large enough (Subsection 4.1)

- We introduce a uniformity test that works for arbitrary $m$ and has sample complexity $O(m^{-1/2})$ (Subsection 4.2)

- In the central DP setting, one can apply a simple reduction approach which consists of drawing $\lceil 1/\epsilon_0 \rceil$ batches of data, and run a non-DP algorithm on one of the batch selected uniformly at random. This approach works well when $m$ is large enough, since the uniformity testing can be solved based on two samples. However, we devise a uniformity test for the central DP setting which has better sample complexity than the reduction approach with data batches for small and medium $m$. This result is presented in Section 5.

- We devise a LDP algorithm as an easy extension of our non-private algorithm, in Section 6.

- We demonstrate the versatility of our algorithm running with large $m$, including $m = 10000$ on synthetic data, and we show that for large $m$ very small deviation form the uniform can be detected with high confidence based on two samples in Section 7.

## 2  Related Work

Testing uniformity is one of the most fundamental problem in computer science. Goldreich and Ron [22] considered first uniformity testing problem as a property testing, however with L2 distance. Paninski [29] came up with a coincidence-based approach that used total variation distance with a sample complexity $O(\sqrt{d}/\epsilon^2)$, where $d$ is the domain size, and it was shown to be optimal by with a restriction that $\epsilon \in \Omega(d^{-1/4})$. The test statistic used by this optimal test is based on number of bins into which just one sample has fallen. In principle, this test can be applied to ranking data, since the test statistic is easy to compute. Nevertheless, the lower bound of this test, which is $\Omega(\sqrt{d}/\epsilon^2)$, suggests that it is not the proper choice of method even for ranking data with small $m$, since $d = m!$.

The result of Paninski [29] was strengthen further by several authors including [12, 11, 3] so as the lower bound does work for general $\epsilon$ and matching algorithms are also provided. The plug-in based tests, for example [11], do estimate the distribution empirically and then it makes a decision by thresholding the total variation distance between the uniform distribution and the empirical estimate. Surprisingly, this test is optimal for all $\epsilon, \delta > 0$. Batu and Canonne [7] considered generalized uniformity testing for an arbitrary distribution with unknown domain size and provided a tester with sample complexity $O(d^{2/3}/\epsilon^6)$. Moreover, Valiant and Valiant [31] showed how to achieve instance optimality, when we test arbitrary discrete distributions for uniformity. However, to adapt any of these tests to our setup is out of question since the size of our domain is $m!$, which is huge, and the test statistic, which is the total variation distance, is already challenging to compute for ranking data. Therefore, we need to come up with a uniformity test that takes advantage of the structure of the alternative hypothesis.

Testing with differential privacy has a solid literature as well. Jayadev Acharya and his colleagues studied uniformity testing with privacy constraint in depth. In [4], the central DP model is analysed and shown that it can be tackled with $\Theta\left(\frac{\sqrt{d}}{\epsilon^2 \epsilon_0} + \frac{\sqrt{d}}{\epsilon \sqrt{\epsilon_0}} + \frac{d^{1/3}}{\epsilon^{4/3} \epsilon_0^{2/3}} + \frac{1}{\epsilon \epsilon_0}\right)$ samples where $\epsilon_0$ is the privacy budget parameter. Local private uniformity test is analyzed in [1] and had found that it can be tackled with $\Theta(k/(\epsilon \cdot \epsilon_0)^2)$ samples. In a recent work [2], uniformity testing in local differential private setting was analysed with a special attention to the impact of the public randomness.

## 3 The Mallows Model, Testing and Differential Privacy

**The Mallows Model.** The *Mallows model* or, more specifically, Mallows $\phi$-distribution is a parametrized, distance-based probability distribution that belongs to the family of exponential distributions $\mathcal{R} = \{\mathcal{M}_{\phi,\pi} \mid \phi \in [0,1], \pi \in \mathbb{S}_m\}$ with probability mass function $p_{\phi,\pi_0}(\pi) = \phi^{d(\pi,\pi_0)}/Z(\phi,\pi_0)$, where $\phi$ and $\pi_0$ are the parameters of the model: $\pi_0 \in \mathbb{S}_m$ is the location parameter also called center ranking and $\phi \in [0,1]$ the spread parameter. Moreover, $d(\cdot,\cdot)$ is a distance metric on permutations, which for our paper will be the Kendall tau distance, that is, the number of discordant item pairs $d_K(\pi,\pi') = \sum_{1 \le i < j \le m} \mathbb{I}\{(\pi(i) - \pi(j))(\pi'(i) - \pi'(j)) < 0\}$. The normalization factor in the definition of the model is equal to $Z(\phi,\pi_0) = \sum_{\pi \in \mathbb{S}_m} p_{\phi,\pi_0}(\pi)$. When the distance metric $d$ is the Kendall tau distance we have the identity $Z(\phi,\pi_0) = Z(\phi) = \prod_{i=1}^{m-1} \sum_{j=0}^{i} \phi^j$.

**Testing and Uniformity Testing.** We assume a parametric family of ranking distribution $\mathcal{R} = \{\mathcal{M}_\theta \mid \theta \in \Omega\}$ where $\Omega$ denotes the set of parameters. The observation consists of $n$ rankings $\mathcal{D}_n = \{\pi_1, \ldots, \pi_n\}$ from a ranking distribution $\mathcal{M}$. The null hypothesis is $H_0 : \mathcal{M} \in \mathcal{R}_0$ where $\mathcal{R}_0 \subset \mathcal{R}$. As an alternative hypothesis, we consider $H_1 : \mathcal{M} \in \mathcal{R}_1(\subset \mathcal{R})$ such that $\mathcal{R}_0 \cap \mathcal{R}_1 = \emptyset$. Then the test is a function $f : \mathbb{S}_m^n \mapsto \{0,1\}$, where 0 corresponds to acceptance, and 1 to rejection. The input of the *tester* (or the *testing algorithm*) is a tolerance parameter $\epsilon > 0$ and a significance parameter $\delta \in (0,1)$. We assume that the tester has sample access of the unknown distribution $\mathcal{M} \in \mathcal{R}$. By definition, an $(\epsilon, \delta)$-tester outputs a sample size $n$ and a test function $f : \mathbb{S}_m^n \mapsto \{0,1\}$ such that, generating $\mathcal{D}_n$ from $\mathcal{M}$, we have the following guaranties for $f$:

1. if the null hypothesis $H_0$ is true, then it outputs reject ($f(\mathcal{D}_n) = 1$) with probability at most $\delta$, i.e. $\mathbb{E}[f(\mathcal{D}_n)] \le \delta$.

2. if $\mathcal{M} \in \mathcal{R}_1$ such that $d_{\text{TV}}(\mathcal{M}, \mathcal{R}_0) > \epsilon$, then it outputs reject ($f(\mathcal{D}_n) = 1$) with probability at least $1 - \delta$, where $d_{\text{TV}}(\mathcal{M}, \mathcal{R}_i) = \inf_{\mathcal{M}' \in \mathcal{R}_i} d_{\text{TV}}(\mathcal{M}, \mathcal{M}')$.

The testing problem at hand is called *uniformity testing*, when the null hypothesis is *simple* and consists of only the uniform distribution over the domain $\mathbb{S}_m$. More concretely, in this study, we assume that the null hypothesis is the uniform model, which is itself a Mallows model with $\phi = 1$, thus $H_0 : \mathcal{M} \in \mathcal{R}_0 = \{\mathcal{M}_{1,\pi_0}\}$, with central ranking an arbitrary $\pi_0 \in \mathbb{S}_m$. The alternative hypothesis class contains those Mallows models that are $\epsilon$-far from the uniform distribution, i.e. $H_1 : \mathcal{M} \in \mathcal{R}_1 = \{\mathcal{M}_{\phi,\pi} : d_{TV}(\mathcal{M}_{1,\pi_0}, \mathcal{M}_{\phi,\pi}) > \epsilon\}$, where $\pi \in \mathbb{S}_m$ is some given central ranking and $\phi \in [0,1)$.

**Differential Privacy.** We also consider uniformity testing under a privacy constraint. We work with two different privacy notions: central and local differential privacy. In case of central differential privacy (CDP), the data is gathered at a so-called *curator*, who is trusted. The curator runs the tester,

which is required to output a response that is not too sensitive to small changes in the input. Small changes can be defined in many different ways for ranking data. We assume that the granularity of the data is whole rankings. With this in hand, we define the notion of CDP as follows.

**Definition 1** (Central DP Property.). *A randomized algorithm $\mathcal{A}$ with domain $\mathbb{S}_m^n$ is $(\epsilon_0, \delta_0)$-differential private if for all $S \subset Range(\mathcal{A})$ and for all pairs of inputs $\mathcal{D} = \{\pi_1, \ldots, \pi_n\}$ and $\mathcal{D}'$ that differs from $\mathcal{D}$ in a single ranking, it holds that*

$$\mathbf{P}(\mathcal{A}(\mathcal{D}) \in S) \le e^{\epsilon_0} \mathbf{P}(\mathcal{A}(\mathcal{D}') \in S) + \delta_0$$

*If $\delta = 0$, the guarantee is called pure differential privacy.*

We will also work with a more appealing privacy notion which is often called locally differential privacy (LDP). LDP is a stronger privacy guaranty than CDP in a sense that the data is required to be privatized before the tester can observe it. More detailed, the data is assumed to be distributed among peers and the peers add noise to the data which noise can be modelled as a conditional distribution, a.k.a. mechanism, $W(\mathbf{z}|\pi)$ for some output space $\mathbf{z} \in \mathcal{Z}$. The tester only observers the privatized data $(\mathbf{z}_1, \ldots, \mathbf{z}_k)$, so in this case it has no access to the original data as it is to be the case in CDP. The notion of LDP therefore can be formalized as a condition on $W(.|.)$

**Definition 2** (Locally differentially private.). *A mechanism $W : \mathbb{S}_m \times \mathcal{Z} \to (0, 1]$ is $\epsilon_0$-locally differentially private if $W$ satisfies*

$$\max_{\mathbf{z} \in \mathcal{Z}} \max_{\pi, \pi' \in \mathbb{S}_m} \log \frac{W(\mathbf{z}|\pi)}{W(\mathbf{z}|\pi')} \le \epsilon_0$$

We assume that each peer has the same mechanism, however there are several setup which assumes that the mechanism can be different for different peers. If public randomness is used by the tester, then the LDP guaranty is extended so as the worst-case log likelihood is computed over the domain of the public random process.

**Definition 3** (Locally differentially private with public randomness). *A set of mechanisms $W_u : \mathbb{S}_m \times \mathcal{Z} \to (0, 1]$ indexed by $U$ which is the domain of the public random process, is $\epsilon_0$-locally differentially private if it holds that*

$$\max_{u \in U} \max_{\mathbf{z} \in \mathcal{Z}} \max_{\pi, \pi' \in \mathbb{S}_m} \log \frac{W_u(\mathbf{z}|\pi)}{W_u(\mathbf{z}|\pi')} \le \epsilon_0 \ .$$

## 4 Non-Private Uniformity Testing

### 4.1 Testing Uniformity of Mallows Models with Two Samples

First, we present a simple algorithm that draws two samples $\pi_1$ and $\pi_2$ and applies only if $\phi_\epsilon$ is bounded away from 1 by $\Omega(1/m)$. The algorithm computes the Kendall tau distance of $\pi_1$ and $\pi_2$. Under the null hypothesis, $d_K(\pi_1, \pi_2) \ge m(m-1)/4 - O(\sqrt{m^3 \ln(1/\delta)})$, with probability at least $1 - \delta$, because the distribution is uniform. Under the alternative hypothesis, $d_K(\pi_1, \pi_2) \le 2\phi_\epsilon m/(1 - \phi_\epsilon) + O(\sqrt{m^3 \ln(1/\delta)})$, with probability at least $1 - \delta$, because the distribution is concentrated around some central ranking. This algorithm is referred to as 2SAMP and is defined in Algorithm 1. We show that if $\phi_\epsilon \le 1 - \Omega(1/m)$, we can distinguish between the two cases by sufficiently large confidence.

**Theorem 4.** *For all $\delta > 0$, if $\phi_\epsilon \le 1 - \frac{8}{m+7-\sqrt{12 \ln(2/\delta)m}}$, then Algorithm 2SAMP, defined in Algorithm 1, uses 2 samples and is an $(\epsilon, \delta)$-uniformity test of Mallows models.*

The proof is deferred to Appendix A. An alternative way of interpreting the guarantee presented above is that for all $m$ and $\phi_\epsilon$, with $m > \frac{9}{1-\phi_\epsilon} - 7$, Algorithm 2SAMP is a $(\epsilon, \delta)$-test for uniformity with significance $\delta$ and power under an alternative model with spread parameter $\phi$ at least $1 - 2e^{-(m+7-\frac{9}{1-\phi_\epsilon})^2/(12m)}$. We should also emphasize that we cannot test uniformity with less than 2 samples, even for very large values of $\epsilon$, since it is impossible to tell whether a single sample is uniformly distributed or not.

The computational complexity of Algorithm 1 is determined by the time required to compute $\phi_\epsilon$. Unfortunately, there is no closed form of $\phi_\epsilon$ as a function of $m$ and $\epsilon$. However, $\phi_\epsilon$ can be computed

---

**Algorithm 1** 2SAMP: Uniformity Test with Two Samples

---

1: **Input:** significance $\delta > 0$, tolerance $\epsilon > 0$
2: Fix any $\pi_0 \in S_m$ and let $\phi_\epsilon = \sup_{\phi \in [0,1]} \{d_{TV}(\mathcal{M}_{1,\pi_0}, \mathcal{M}_{\phi,\pi_0}) > \epsilon\}$
3: **if** $\phi_\epsilon > 1 - \frac{8}{m+7-\sqrt{12\ln(2/\delta)m}}$ **then**
4:     Output $\perp$                                      ▷ $\phi_\epsilon$ too close to 1
5: Take 2 samples which are denoted by $\pi_1, \pi_2 \in S_m$
6: **If** $d_K(\pi_1, \pi_2) > \frac{m(m-1)}{4} - \sqrt{\frac{m^3 \ln(1/\delta)}{12}}$ **Then** Accept **Else** Reject

---

efficiently, using dynamic programming and binary search. We first observe that the total variation distance of a Mallows model to the uniform distribution can be computed in $\Theta(m^2)$ time, using $\sum_{i=0}^{m(m-1)/2} \left| \frac{1}{m!} - \frac{\phi^i}{Z(\phi)} \right| \mathrm{Mah}(i, m)$, where $\mathrm{Mah}(i, m)$ denotes the $i$-th Mahonian number[1] of order $m$ (i.e., the number of permutations with $m$ items at Kendal tau distance $i$ to the identity permutation). The Mahonian numbers of order $m$ can be computed in $\Theta(m^2)$ time, using dynamic programming, while $Z(\phi)$ has a closed form [21]. Furthermore, the total variation distance of a Mallows model to the uniform distribution decreases as $\phi$ increases from 0 to 1. Hence, $\phi_\epsilon$ can be computed efficiently using binary search and the total variation distance computation above.

## 4.2 General Uniformity Testing Algorithm

Algorithm 2SAMP works only when $\phi_\epsilon \leq 1 - \Omega(1/m)$, or equivalently, assuming a fixed $\phi_\epsilon$, when $m$ is large enough. Hence, we present Algorithm 2, which tests uniformity for arbitrary $\phi_\epsilon$ and $m$. The idea is to consider the relative positions of $m/2$ disjoint random pairs of items in $k = \Theta\left(\frac{1}{\mu^2}\sqrt{\frac{\ln(1/\delta)}{m}}\right)$ samples, where $\mu = \frac{1-\phi_\epsilon^{m/8}}{1+\phi_\epsilon^{m/8}}$. Under the alternative hypothesis $H_1$ with any fixed central ranking $\pi^*$, for any pair of items $i$ and $j$, with $\pi^*(j) = \pi^*(i) \geq m/8$, $\mu$ is a lower bound on the bias towards observing $i$ before $j$ in a random ranking from $H_1$ (if $m$ is large enough, a random item pairing results in $m_1 \approx m/8$ such item pairs, with high probability). We define the random variables $X_{i(i+1)}^\ell$ to be 1, if $i$ precedes $i+1$ in sample $\pi_\ell$, and $-1$, otherwise. $Y_{i(i+1)} = \left(\frac{1}{\sqrt{k}} \sum_{\ell=1}^k X_{i(i+1)}^\ell\right)^2$ accounts for the deviation of the pair $i$ and $i+1$ from uniformity. Under the null hypothesis, the expectation and the variance of $Y_{i(i+1)}$ are $O(1)$. Under the alternative hypothesis, $\mathbb{E}[Y_{i(i+1)}] = \Omega(k\mu^2)$ and $\mathbb{V}[Y_{i(i+1)}] = O(k\mu^2)$. Moreover, the random variables $Y_{12}, Y_{34}, \cdots, Y_{(m-1)m}$ are mutually independent, because they concern the relative positions of disjoint item pairs in the samples. Therefore, $Y = Y_{12} + Y_{34} + \cdots + Y_{(m-1)m}$ should be $O(m)$, under the null hypothesis, and $\Omega(mk\mu^2)$, under the alternative hypothesis. The following is proven in Appendix B and shows that we can distinguish between the two cases with adequate confidence.

---

**Algorithm 2** Uniformity Test (UNIF)

---

1: **Input:** significance $\delta > 0$, tolerance $\epsilon > 0$
2: Let $\pi_0 \in S_m$ be chosen uniformly at random and renumber the items so that $\pi_0 = (1, \ldots, m)$
3: Let $\phi_\epsilon = \sup_{\phi \in [0,1]} \{d_{TV}(\mathcal{M}_{1,\pi_0}, \mathcal{M}_{\phi,\pi_0}) > \epsilon\}$     ▷ $\phi_\epsilon$ does not depend on $\pi_0$
4: Let $\mu = \frac{1-\phi_\epsilon^{m/8}}{1+\phi_\epsilon^{m/8}}$ and $m_1 = m/8 - \sqrt{m\ln(2/\delta)/16}$.
5: Take $k = 1 + \left\lceil \frac{1}{\mu^2}\left(\frac{12m\ln(2/\delta)}{m_1^2} + 10\sqrt{\frac{m\ln(2/\delta)}{m_1^2}}\right)\right\rceil$ samples $\mathcal{D}_k$.
6: Let $\pi_1, \ldots, \pi_k \in S_m$ denote the samples
7: Let $X_{i(i+1)}^\ell = 1$, if $i \succ_{\pi_\ell} i+1$, and $-1$ otherwise.
8: Let $Y_{i(i+1)} = \left(\frac{1}{\sqrt{k}} \sum_{\ell=1}^k X_{i(i+1)}^\ell\right)^2$ for all item pairs $i(i+1)$
9: Let $Y = Y_{12} + Y_{34} + \cdots + Y_{(m-1)m}$
10: **If** $Y < m/2 + 2\sqrt{m\ln(1/\delta)}$ **Then** Accept **Else** Reject

---

---
[1] https://oeis.org/A008302

**Theorem 5.** *For all $\delta, \epsilon > 0$, Algorithm 2 with*

$$k = \Theta\left(\frac{1}{\mu^2}\sqrt{\frac{\ln(1/\delta)}{m}}\right)$$

*samples, where $\mu = \frac{1-\phi_\epsilon^{m/8}}{1+\phi_\epsilon^{m/8}}$, is an $(\epsilon, \delta)$-finite confidence uniformity test of Mallows models.*

Regarding the dependence of $\mu$ on $m$ and $\epsilon$, we observe that if $\epsilon = \Omega(1/\sqrt{m})$, then $\phi_\epsilon \geq 1 - \frac{c\epsilon}{m} \leq e^{-c\epsilon/m}$, for some constant $c > 0$. Then, $\mu \geq \frac{1-e^{-c\epsilon/8}}{1+e^{-c\epsilon/8}}$, which does not depend on $m$. Hence, unless $\epsilon$ is extremely small, i.e., if $\epsilon = o(1/\sqrt{m})$, $\mu$ is constant in $m$ and the sample complexity of Algorithm 2 is $O(m^{-1/2})$.

Algorithm 2 exploits a tradeoff between the bias $\mu$ towards the right order of $i$ and $i+1$ and the values of $\phi_\epsilon$ and $m$. If $m$ is so small that $m_1 = m/8 - \sqrt{m\ln(2/\delta)/16}$ becomes much smaller than $m$, the gain in the sample complexity due to raising $\phi_\epsilon$ to $m/8$ in $1/\mu^2$ may be counterbalanced by the increase in sample complexity due to $m/m_1^2$. Then, we can apply the same analysis with $\mu' = \frac{1-\phi_\epsilon}{1+\phi_\epsilon}$, essentially regarding all $i(i+1)$ pairs as consisting of consecutive elements in $\pi_0$. Then, under the alternative hypothesis, $\mathbb{E}[Y] = m(1 + (k-1)\mu'^2)/2$, which gives a sample complexity of

$$k = 1 + \left\lceil \frac{(1+\phi_\epsilon)^2}{(1-\phi_\epsilon)^2}\left(\frac{48\ln(1/\delta)}{m} + 16\sqrt{\frac{\ln(1/\delta)}{m}}\right)\right\rceil$$

for Algorithm 2. More generally, if $m$ is relatively small (so small that $m_1$ becomes negligible), a more careful analysis of Algorithm 2 is possible. Then, in the proof of Theorem 5, we can optimize the sample complexity by trading off the exponent of $\phi_\epsilon$ in $\mu$ against the size of $m_1$. As a result, the sample complexity of Algorithm 2 is upper bounded by a family of functions that have the same form as (5) in the proof of Theorem 5, Appendix B, but use different values of $m_1$ and $\mu$, where we can increase $m_1$ from $m/8$ up to $m$, by subsequently decreasing the exponent of $\phi_\epsilon$ in $\mu$ from $m/8$ to 1.

## 5 Uniformity Test with Central Differential Privacy

The most natural approach to make a testing algorithm differentially private is to add Laplace noise to the test statistic. The variance of the noise should be proportional to the sensitivity of the test statistic, which ensures that small change in the input data does not change drastically the output of the tester. This intuition is formalized in [17, Theorem 3.6]. Accordingly, one can add Laplace noise to the test statistic $Y$ which is computed in line 9 of Algorithm 2. What remains is to compute the sensitivity of $Y$ with respect to the input. Assume that we are given two datasets $\mathcal{D}_k$ and $\mathcal{D}'_k$ that consist of $k$ rankings and differ from each other in one ranking. Then, the sensitivity of the test statistic computed for $\mathcal{D}_k$ and $\mathcal{D}'_k$ is $|Y(\mathcal{D}_k) - Y(\mathcal{D}'_k)| \leq 2m$. This calculation is presented in Claim 10 in the Appendix. Therefore, if we add a Laplace noise $Z \sim \mathbf{Lap}\,(2m/\epsilon_0)$ to $Y$ in line 9 of Algorithm 2, our algorithm becomes $(\epsilon_0, 0)$-differentially private. However, note that $\mathbb{V}\,[\mathbf{Lap}\,(2m/\epsilon_0)] = 8\,(m/\epsilon_0)^2$, which is much larger than the variance $\mathbb{V}\,[Y] \leq m$ of the test statistic of Algorithm 2. Consequently, this noise results in a large increase of sample complexity. For sake of completeness, this approach is presented in Algorithm 4, in Appendix C. The only change with respect to (non-private) Algorithm 2 is that there is an extra Laplace noise added to the test statistic in line 7 and the rejection threshold has updated accordingly.

The sample complexity of Algorithm 2 is $O\big(1/\mu^2\sqrt{\ln(1/\delta)/m}\big)$, whereas the privatized test requires $O\left(m\sqrt{\ln 1/\delta}/(\mu^2\epsilon_0)\right)$. This means that the naive approach results in an increase in the sample complexity by a factor of $m^{3/2}/\epsilon_0$. Next, we try to bring this variance down with a truncation technique applied to Laplace noise.

In Claim 10, it becomes apparent that the sensitivity of the test statistic $Y$ is proportional to the value of the following sum of independent Rademacher random variables $\left|\frac{4}{k}\sum_{i=1}^{m/2}\sum_{\ell=1}^{k} X_{(2i-1),2i}^\ell\right|$, which can be up to $2m$. To control the variance (and the sample complexity) of the privatized test statistic, we deal with the case where the value of this sum is $\Omega(\sqrt{m/k})$ separately. This is the

intuition behind Algorithm 3, which is inspired by [10, Algorithm 1], and distinguishes between different cases. In particular, if the value of the Laplace noise happens to be high, Algorithm 3 returns a random response, while if $\left|\frac{4}{k}\sum_{i=1}^{m/2}\sum_{\ell=1}^{k}X_{(2i-1),2i}^{\ell}\right| = \Omega(\sqrt{m/k})$, the algorithm rejects. Otherwise, the algorithms performs a randomized version of the uniformity test called UNIF, defined in Algorithm 2. In this case, we show that the sensitivity of the test statistic (and thus, the sample complexity) can be upper bounded reasonably well. We refer to the DP algorithm as TRUNC, which is defined in Algorithm 3.

---

**Algorithm 3** Central DP Uniformity Test (TRUN)

1: **Input:** significance $\delta > 0$, tolerance $\epsilon > 0$, DP parameter $\epsilon_0$
2: Sample $Z \sim \mathbf{Lap}\left(\frac{4}{k\delta\epsilon_0}\right)$
3: **if** $|Z| > \frac{4}{k\delta\epsilon_0}\ln\frac{1}{\delta}$ **then**     ▷ True with probability at most $\delta$
4:     Output Accept or Reject with equal probability
5: Let $\pi_0 \in S_m$ be chosen uniformly at random and renumber the items so that $\pi_0 = (1,\ldots,m)$
6: Let $\phi_\epsilon = \sup_{\phi\in[0,1]}\{d_{TV}(\mathcal{M}_{1,\pi_0},\mathcal{M}_{\phi,\pi_0}) > \epsilon\}$
7: Let $\mu = \frac{1-\phi_\epsilon^{m/8}}{1+\phi_\epsilon^{m/8}}$ and $m_1 = m/8 - \sqrt{m\ln(2/\delta)/16}$.
8: Take $k$ samples $\mathcal{D}_k = \{\pi_1,\ldots,\pi_k\}$, where $k$ is as in Theorem 6.
9: Let $X_{i(i+1)}^{\ell} = 1$, if $i \succ_{\pi_\ell} i+1$, and $-1$ otherwise.
10: **if** $\left|\frac{2}{mk}\sum_{i=1}^{m/2}\sum_{\ell=1}^{k}X_{(2i-1),2i}^{\ell} + Z\right| > \sqrt{\frac{2\ln(2/\delta)}{mk}} + \frac{4}{\delta k\epsilon_0}\ln\frac{1}{\delta}$ **then**
11:     Reject
12: **else**
13:     Let $Y_{i(i+1)} = \left(\frac{1}{\sqrt{k}}\sum_{\ell=1}^{k}X_{i(i+1)}^{\ell}\right)^2$ for all item pairs $i(i+1)$
14:     Let $Y = Y_{12} + Y_{34} + \cdots + Y_{(m-1)m}$
15:     $B \sim \mathbf{Bernoulli}(p)$ where $p = \min\left\{1,\frac{16(Y-m/2)}{m\mu^2(k-1)}\right\}$
16:     **If** B=1 **Then** Accept **Else** Reject

---

**Theorem 6.** *For all $\delta,\epsilon,\epsilon_0 > 0$, Algorithm 3 with $\mu = \frac{1-\phi_\epsilon^{m/8}}{1+\phi_\epsilon^{m/8}}$ and*

$$k = \Theta\left(\max\left\{\frac{1}{\mu^2}\left(\frac{1}{\delta^{3/2}\sqrt{m}} + \frac{1}{\delta^3 m}\right), \frac{\ln^{1/3}(1/\delta)}{\mu^{4/3}m^{1/3}\epsilon_0^{2/3}\delta^{2/3}}, \frac{\sqrt{\ln(1/\delta)}}{\mu\epsilon_0\delta}\right\}\right)$$

*samples is an $(\epsilon_0,0)$-differentially private $(\epsilon,\delta)$-uniformity test of Mallows models.*

Theorem 6 has some interesting consequences. First, privacy comes for free in some parameter regime since the first term of the sample complexity, which is dominant if $\phi_\epsilon$ is close to 1 and $m$ is not so large, does not depend on the privacy parameter $\epsilon_0$. Second, the sample complexity of TRUNC is worse than the simple *bucketing approach* for other parameter regimes. The bucketing approach is a folklore result to convert non-private algorithms to private ones. It consists of running a non-private algorithm on $\lceil 1/\epsilon_0\rceil$ number of data batches in parallel, and return one of the outcomes selected uniformly at random. This approach is $(\epsilon_0,0)$-differentially private and has sample complexity $\lceil 1/\epsilon_0\rceil$ times that of the non-private algorithm (see e.g., [10, Theorem 2] for the precise reduction approach). Combining UNIF with the bucketing approach, we obtain a $(\epsilon_0,0)$-differentially private algorithm with sample complexity $O\left(\max\{1/(\mu^2\epsilon_0)\sqrt{\ln(1/\delta)/m}, 2/\epsilon_0\}\right)$. The bucketing approach is very efficient in our uniformity testing setup, because the sample complexity is of order $m^{-1/2}$ for the non-private algorithm. So, the bucketing approach requires $O(1/\epsilon_0)$ samples, if $m$ is large enough. On the other hand, TRUNC can be superior when $\phi_\epsilon$ is close to 1 and $m$ is not so large, in which case the first term of Theorem 6 becomes dominant, as our experimental evidence also justifies.

## 6 Uniformity Test with Local Differential Privacy (LDP)

Algorithm 2 can easily extended so as it satisfies the LDP constraint, since it extracts a binary sequence from the rankings in Line 7 of Algorithm 2. Adding randomized response (RR) to this bit

sequence componentwisely results in an simple LDP uniformity testing algorithm. Let us denote the conditional probability of RR by $W(.|.)$, for which $W(-1|-1) = W(+1|+1) = \frac{e^\gamma}{e^\gamma+1}$ and $W(-1|+1) = W(+1|-1) = \frac{1}{e^\gamma+1}$. As a consequence, if all $X^\ell_{i(i+1)}$ are passed through a channel $W(.|.)$ with $\gamma = 2\epsilon_0/m$, then the LDP guarantee is satisfied. To see that, we consider two rankings $\pi$ and $\pi'$, for which $f_{\pi_0}(\pi) = (-1, \ldots, -1)$ and $f_{\pi_0}(\pi') = (1, \ldots, 1)$ for a fixed $\pi_0$. Then

$$\log \frac{\mathbf{P}((1,\ldots,1)|\pi)}{\mathbf{P}((1,\ldots,1)|\pi')} = \frac{m}{2} \log \frac{\frac{e^\gamma}{e^\gamma+1}}{\frac{1}{e^\gamma+1}} = \frac{m\gamma}{2} = \epsilon_0$$

We present our LDP uniformity testing algorithm so as it requires public/shared randomness in the form of a random ranking that is sent to each peer beforehand. The curator algorithm is defined in Algorithm 5, and the peer algorithms in Algorithm 6, in Appendix F. The LDP algorithm is based on Algorithm 2 and uses the same test statistic. If public randomness is available, we can implement Algorithm 2 with random item pairing and $\mu_0 = \frac{1-\phi_\epsilon^{m/8}}{1+\phi_\epsilon^{m/8}} \cdot \frac{e^\gamma-1}{e^\gamma+1}$, which leads to an improved sample complexity, if $\phi_\epsilon$ is close to 1 and $m$ is relatively large. If only private randomness is available, we implement Algorithm 2 with a fixed item pairing and $\mu'_0 = \frac{1-\phi_\epsilon}{1+\phi_\epsilon} \cdot \frac{e^\gamma-1}{e^\gamma+1}$ (see also the discussion after Theorem 5 on how the choice of item pairing affects the value of $\mu$ and the sample complexity). The analysis is essentially identical to the proof of Theorem 5, since the mean value and the variance of the test statistic are $m/2$ and at most $m$, respectively, under $H_0$, and are given by the same functions of $\mu$ (or $\mu'_0$), under $H_1$. The only essential difference is the decrease of $\mu$ (or $\mu'_0$) by a factor of $\frac{e^\gamma-1}{e^\gamma+1}$, to account for the randomized response. The discussion above is summarized by the following:

**Theorem 7.** *For all $\delta, \epsilon, \epsilon_0 > 0$ and $\gamma = 2\epsilon_0/m$, Algorithm 5 and Algorithm 6 form an $(\epsilon, \delta)$-finite confidence uniformity test and $(\epsilon_0, 0)$-locally differentially private for Mallows modesls with sample complexity:*

- $k = \Theta\left(\frac{1}{\mu_0^2}\sqrt{\frac{\ln(1/\delta)}{m}}\right)$, *where* $\mu_0 = \frac{1-\phi_\epsilon^{m/8}}{1+\phi_\epsilon^{m/8}} \cdot \frac{e^\gamma-1}{e^\gamma+1}$, *if $m$-bit public randomness is used and a random ranking is sent by the curator algorithm to each peer.*

- $k = \Theta\left(\frac{1}{\mu_0'^2}\sqrt{\frac{\ln(1/\delta)}{m}}\right)$. *where* $\mu'_0 = \frac{1-\phi_\epsilon}{1+\phi_\epsilon} \cdot \frac{e^\gamma-1}{e^\gamma+1}$, *if only private randomness is used.*

Since for $\gamma = 2\epsilon_0/m$, $\frac{e^\gamma-1}{e^\gamma+1} \approx \frac{2\epsilon_0}{m}$, the dependence of the sample complexity of our LDP algorithm on $\epsilon_0$ is $\Theta(1/\epsilon_0^2)$ and on $m$ is $\Theta(m^{3/2})$.

## 7 Experiments

We shall present synthetic experiments to assess the performance of the proposed tests. We assess the power of these tests which is the probability of the rejection for various spread parameter $\phi$. Every testing algorithm we presented has a tolerance parameter $\epsilon$ and significance $\delta$. We used $\delta = 0.05$ in every case. The tolerance parameter $\epsilon$ does have impact only on the sample size of the testing algorithms. Instead of setting $\epsilon$ to a certain value, we plotted the power of the algorithms with various sample size. In this way, we could compare the performance of the testing algorithms based on the same number of samples as input. Each result we report here are computed based on 1000 repetitions. The central ranking of each model which the random samples are generated from, is selected uniformly at random in each each run independently.

### 7.1 Uniformity Testing Based on Two Samples

In the first set of experiments, we compare the uniformity testing algorithm based on two samples, called 2SAMP which is defined in Algorithm 1 and the more general algorithm, called UNIF which is defined in Algorithm 2 running with two rankings as input. We assess the power of these tests which is the probability of the rejection for various spread parameter $\phi$. The results are plotted in Figure 1. The 2SAMP algorithm does work already for $m = 100$ and it consistently outperforms UNIF based on two rankings. This can explained by the fact that 2SAMP computes pairwise statistic, which is the Kendall distance, based on each pair of items, whereas UNIF takes into account only the independent pairs. It is worth to emphasize that these tests can detect very small deviation from

the uniform for large $m$. More concretely, it is detected with zero error, i.e. power is equal to 1, when the spread parameter deviates from 1 with a margin of $2 \times 10^{-5}$ in case of $m = 10000$.

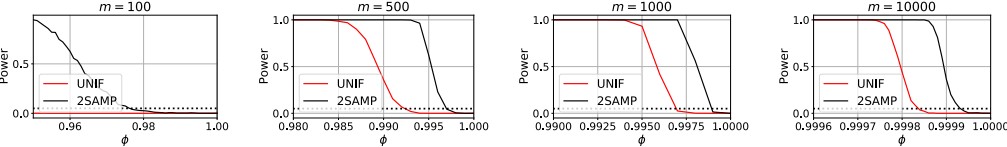

Figure 1: Power function of 2SAMP introduced in Subsection 4.1 and UNIF introduced in Subsection 4.2 with various parameters of the alternative model. The spread parameter of the underlying Mallows model is shown on the x-axis.

## 7.2 General Uniformity Testing with Arbitrary $m$

Testing uniformity was one of the motivation of Mallows when he came up with his model (see Section 11 in [25]). Mallows assumed that the central ranking is known and proposed a asymptotic test based on the normal approximation of the sufficient statistic of his model. Even if there is no guaranty of how good this normal approximation is, it seems reasonable when the central ranking is fixed, since the distribution of the sufficient statistics is symmetric due to the fact that Mahonian numbers are symmetric, i.e. $N_k = N_{m(m-1)/2-k}$ where $N_k$ is the $k$th Mahonian number of order $m$. One can compute the mean and variance of sufficient statistic based on [20] as

$$\mathbb{E}_{\pi \sim \mathcal{M}_{1,\pi_0}} \left[ T_{\pi_0}(\pi) \right] = \sum_{i=0}^{M} \frac{i N_i}{m!} = \frac{m(m-1)}{4}$$

and

$$\mathbb{V}_{\pi \sim \mathcal{M}_{1,\pi_0}} \left[ T_{\pi_0}(\pi) \right] = \frac{m(2m+5)(m-1)}{72}.$$

This approximate solution is easy to use, since uniformity testing boils down to testing equality of expectation of normal distributions with known variance. Here we consider a more general testing problem where we do not assume that the central ranking is known and fixed. We refer to this approach as Mallows approximate test (MA). In this test, the normal approximation seems not so accurate as Figure 2 shows. The power of the test converges to 1 very slowly as $\phi$ is getting far from zero. Algorithm UNIF achieves a power that is close to one much faster.

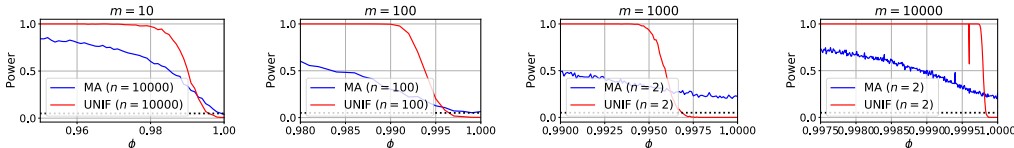

Figure 2: Power function of MA and UNIF algorithms with various number of items $m \in \{10, 100, 1000, 10000\}$ and various sample size $n \in \{10000, 100, 2, 2\}$, respectively.

## 7.3 Uniformity Testing with Privacy

In this set of experiments, we compare the performance of the presented DP algorithms including the following three of them: (1) bucketed UNIF which consists of running the UNIF algorithm on $\lceil 1/\epsilon_0 \rceil$ batches of data indpendently and take the output of one of the runs uniformly at random. We refer to this approach as BUNIF. (2) We run the DP algorithm with truncated Laplace noise that is defined in Algorithm 3 which we refer to as TRUNC. (3) We run also the locally differentially private algorithm which is defined in Subsection 6. We set $\epsilon_0 = 0.33$ thus the BUNIF power curve corresponds to the UNIF power curve with $1/3$ of the sample complexity.

The power curves of the algorithms are shown in Figure 3. Note that the TRUNC does outperforms BUNIF when $\phi$ is close to 1, since in that case $\mu$ is very close to 1 and the first term is the dominating

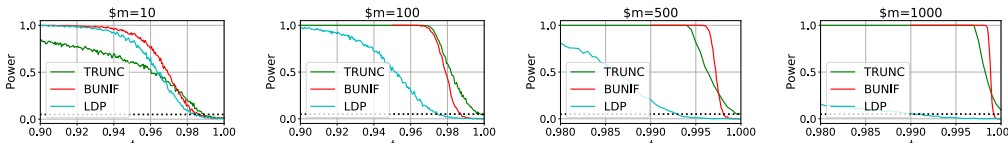

Figure 3: Power function of BUNIF, TRUNC and LDP algorithms with various number of items $m \in \{10, 100, 500, 1000\}$ and various sample size. We set the sample size for central DP algorithms, i.e. BUNIF and TRUNC, to $n \in \{10000, 1000, 1000, 100\}$, respectively. In case of LDP, the sample size was set to $n = \{100k, 200k, 300k, 300k\}$. The privacy budget $\epsilon_0$ was set to $1/3$.

one in Theorem 6. Nevertheless, the power of BUNIF algorithm converges faster to 1 when the underlying model getting farther from the uniformity. In general, the DP algorithms requires much more samples than the non-private algorithms. This difference is much more pronounced in case of LDP. However, local differential privacy is well known to provide some of the strongest privacy guarantees as it is impossible for an adversary to know the true ranking from the output of the channel.

## 8 Conclusion and Future Work

We introduced uniformity testing algorithm with a sample complexity upper bound of order $O(1/\sqrt{m})$ for ranking data when the alternative model class is constituted by the single parameter Mallows model. The proposed methods can work based on 2 samples, when $m$ is large enough. We also devised testing algorithms in the central and local differential privacy framework. We demonstrated the versatility of these testers on synthetic data. We found that they are scalable, since they could handle large $m$ including $m = 10000$, and are able to detect non-uniformity with very small error, i.e. $1 - \phi \approx 10^{-5}$ difference could be detected with zero error based on two samples when $m = 10000$.

One open question is to provide lower bounds on the sample complexity of uniformity testing of Mallows models. It turned out that this is a very challenging problem in the non-private case since, for example, using information theoretic lower bounding technique based on LeCam's theorem, one needs to upper bound KL divergence $\mathrm{KL}(\mathcal{M}_{\phi_\epsilon, \pi_0}, \mathcal{M}_{1,\pi_0}) = \ln^2 \phi_\epsilon \nabla^2 \ln Z(\xi)$ where $\xi \in [\ln \phi_\epsilon, 0]$. For doing so, one needs to have an upper bound for the difference of $\phi_\epsilon$ from 1 in terms of $\epsilon$ and $m$. Note that $\phi_\epsilon$ depends on the total variation distance, thus it seems unavoidable to get a bound for $Z(\phi_\epsilon)$ as well, which is a very hard nut to crack. So we leave this lower bound as future work. Nevertheless, note that with $m = 2$, the single parameter Mallows class includes the class of Bernoulli distributions, thus the lower bounds that are devised for private uniformity testing of Bernoulli distributions also apply to private uniformity testing of Mallows models. More concretely, [5] showed that testing Bernoulli with parameter in a central DP setting cannot be done using $o(1/\epsilon_0)$ which implies that our central DP algorithm has optimal dependency on the privacy budget parameter. On the other hand, [6] showed that in the LDP setting, a lower bound on the sample complexity of testing is $\Omega(1/\epsilon_0^2)$. Theorem 7 matches this lower bound when $m = 2$, since then $e^\gamma - 1 \approx \epsilon_0$ and the sample complexity bound is $O(1/\epsilon_0^2)$ in this case. Lastly, a natural extension of our work is to consider more fine grained atom in the privacy setting, namely one might want to protect each pair of the data instead of whole rankings.

## Acknowledgments and Disclosure of Funding

Dimitris Fotakis is supported by the Hellenic Foundation for Research and Innovation (H.F.R.I.) under the "First Call for H.F.R.I. Research Projects to support Faculty members and Researchers and the procurement of high-cost research equipment grant", project BALSAM, HFRI-FM17-1424.

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
