# OpenReview forum: "Private and Non-private Uniformity Testing for Ranking Data"
_NeurIPS.cc/2021/Conference — NeurIPS 2021 Poster_

### Official Review · Reviewer_G2iJ · 2021-07-03

**Rating:** 7
**Confidence:** 4

**Summary:**


The authors study the problem of differentially private (DP) uniformity testing for rankings:
are the rankings uniformly and randomly generated (under the null hypothesis) or are the rankings
generated in such a way that the distribution of rankings are $\epsilon$ far from the rankings generated under
the uniform distribution.
They present two main algorithms: 2SAMP (works in certain paramter regimes and uses two samples) and UNIF.
In Section 7, they present simulations on synthetic data to validate their methods.

**Ethics Review Area:**

["I don’t know"]

**Limitations And Societal Impact:**

They address the limitations of 2SAMP compared to UNIF and vice versa.  Also, in the conclusion section, they mention some open questions.

**Main Review:**

Their results are obtained under the Mallows model, which is widely used in the ML rankings literature.
The model has two main inputs: $\pi_0$ and $\phi\in[0, 1]$. $\pi_0$ is a central ranking and
$\phi$ is the "spread" around this central ranking.

Algorithm 1 (2SAMP) is a uniformity tester that uses two samples and works only when $\phi\leq 1 - \Omega(1/m)$,
where $m$ is the number of items to be ranked.
Algorithm 2 (UNIF), on the other hand, works on any arbitrary $\phi$.

In Section 7, Figures 1, 2, and 3 serve to experimentally validate their methods, algorithms, and assumptions.

I believe that their work provides novel applications (to uniformity testing) of known methods in the DP literature. They provide both central and local DP algorithms for uniformity testing.

Minor Comments
==============

- Use of $\epsilon$s and $\delta$:
I think $\epsilon_0$ is used as the privacy parameter and $\epsilon$ is used for how
far from uniform the distribution of rankings is, under the alternative hypothesis.
And $\delta$s are used for significance? This might be confusing. [response acknowledged]

- Line 202 states that the variance of the Laplace is $8b^2$ for scale b. Shouldn't this be $2b^2$? [response acknowledged]



**Time Spent Reviewing:**

approximately 3 hours

---

> ### Author Response · Authors · 2021-08-09
> **Response to Reviewer G2iJ**
>
> Thanks for the positive feedback.
> * We used $\epsilon$ and $\delta$ to denote the testing tolerance and significance, respectively, and $\epsilon_0$ and $\delta_0$ for the DP parameters. Thank you for pointing out this, these parameters are not used consistently everywhere in the paper. We will make sure to fix it.
> * Regarding line 202: it should be $8(m/\epsilon_2)^2$ as the reviewer pointed out.

---

### Official Review · Reviewer_Czjv · 2021-07-16

**Rating:** 7
**Confidence:** 4

**Summary:**

This paper studies uniformity testing for ranking data in which the alternative class is restricted to the Mallows model. In the non-private setting, the authors first give a uniformity tester that only needs two samples for large enough m (where m is the number of items over which the ranking is considered), then the authors give a general tester that works for arbitrary m. In the private setting, the authors give differentially-private uniformity testers for both the centralized and local models of DP. They provide synthetic experiments to support their theoretical results.


**Limitations And Societal Impact:**

Yes/

**Main Review:**

Testing uniformity of data is a well-studied direction in property testing. There has also been recent work on testing uniformity under differential privacy constraints. In this work the authors focus on testing if a distribution is the uniform distribution over all permutations on m elements (called rankings) or epsilon far from this distribution in total variation distance, both in the standard setting and under central and local differential privacy. The collection of
distributions that are far are considered in the Mallows model: a distribution over permutations is parameterized by parameters phi in [0,1] (the spread) and a fixed permutation pi0 (the center), and the probability mass function assigns mass to permutation pi that is a normalization of phi^{d(pi, pi0), where d is the Kental tau distance of two permutations, thus counting the number of inversions.

The paper starts with a simple test that picks two rankings from the given distribution, and correctly accepts/rejects as long as the parameter phi is large enough (>1-\Omega(m)).

Leveraging this simple test, the authors obtain a general test. The idea is to draw samples and estimate the variance of the relative ordering of m/2 pairs of items. They show that this is enough to distinguish between the two cases.

Here, please discuss for what parameters the sample complexity is interesting. In many places (eg, abstract and conclusions) where the phi’s are suppressed your sample complexity is below 1! (m^{-½}). Please fix these typos everywhere and add a discussion about the tradeoffs between the parameters of the model and m.

To get the DP version, they first observe that by adding Laplace noise proportional to the sensitivity of the test statistic the query complexity would blow up by another factor of m^{3/2}/ To bring this down, the authors carefully add Laplacian noise to the previous algorithm only when the variance is well-behaved, namely it’s not too small and not too large. Achieving this step uses some ideas from [9]. Finally, achieving LDP appears to be much easier now, as one can add some randomized response to the output of the classical algorithm.

Originality

Although uniformity testing has a vast literature, uniformity testing for ranking data in which the alternative class is restricted to the Mallows model has not been well-studied, so the results presented are novel.

Quality
The paper is generally well-written, with full definitions and clear exposition. I have not verified the technical proofs in the supplemental material.

Clarity
There are minor typos and grammatical errors sprinkled across the entire submission. For e.g., Line 88: ``Local private uniformity test is analyzed in [1] and had found that it can…’’ Please also see the comment above about the typos in the testing complexity.


Significance
The paper would be interesting to the property testing, differential privacy and machine learning communities.




**Time Spent Reviewing:**

1

---

> ### Author Response · Authors · 2021-08-09
> **Response to Reviewer Czjv**
>
> Thank you for the positive feedback.
>
> Regarding the dependence on the parameters: Our sample complexity bound is parametrized by $\mu$ which depends on both $\epsilon$ and $m$ in general. Unfortunately, no closed form of $\phi_\epsilon$ as a function of $\epsilon$ and $m$ is known (and does not seem manageable to obtain such a good approximation for all values of $\epsilon$), which would allow us to fully quantify the dependence. But, if $\epsilon = \Omega(1/\sqrt{m})$, then $\phi_\epsilon \leq 1 - \frac{c \epsilon}{m} \leq e^{-c \epsilon / m}$, for some constant $c > 0$. Then, $\mu \geq \frac{1-e^{-c \epsilon /8}}{1+e^{-c \epsilon/8}}$, which does not depend on $m$. Thus, unless $\epsilon$ is extremely small, i.e. unless $\epsilon = o(1/\sqrt{m})$, $\mu$ is indeed constant in $m$. That is why we decided to present our results in this way. We will add more discussion on how the sample complexity depends on various parameters.

---

> > ### Comment · Reviewer_Czjv · 2021-08-17
> > **response**
> >
> > I am happy with the way the authors addressed my comments.

---

### Official Review · Reviewer_r9MB · 2021-07-19

**Rating:** 6
**Confidence:** 4

**Summary:**


The paper considers the problem of uniformity testing for permutations against the Mallows model with an unknown central ranking. More precisely, given random samples  $\mathcal D_n=\{\pi_1, \pi_2, \ldots \pi_n\}$  from $S_m$ (the collection of all permutations of size $m$), the authors determine the sample complexity for testing whether $\mathcal D_n$ is generated from the uniform distribution on $S_m$ versus whether $\mathcal D_n$ is generated from some Mallows model with total variation distance $\epsilon$ from the uniform distribution. They provide an algorithm for this testing problem with sample complexity $O(1/\sqrt m)$.  The authors then study this problem under privacy constraints. In particular, they provide algorithms  for uniformity testing in the Mallows model under central as well as local differential privacy constraints. The results are validated by simulation experiments.


**Limitations And Societal Impact:**



As mentioned above, it is sometimes unclear how some of the quantities depend on the sample complexity. For example, what is the dependence on $\epsilon$ in the results? Is this dependence expected to optimal?

Also, it might be better to assume that $\delta$ is fixed. It is unlikely that the dependence on $\delta$ is optimal and it would significantly clean up the sample complexity in the private setting.


**Main Review:**



The problem studied in the paper is interesting and relevant. However, the paper often omits important details and it is difficult to parse the  sample complexities. It is unclear what the dependencies on the various parameters are and how close they are to the optimal, which makes it difficult to appreciate the technical contributions of the paper. Moreover, the main ideas of the papers are packed in to the algorithms and it is not clear how some of the quantities are computed.


Specific Comments:


I am not sure if this is because of space constraints, but a few important recent references on uniformity testing problems are missing. For example:

G. Valiant and P. Valiant, An automatic inequality prover and instance optimal identity testing,  SIAM J. Comput., 46(1):429--455,  2017.

T. Batu and C. Canonne, Generalized uniformity testing, FOCS, 2017.


Page 3, Line 120: In the definition of the class $\mathcal R_1$, it would be good to mention that $\phi$ can vary over $[0, 1)$ and the central permutation $\pi_0$ over $S_m$.


Page 4, Algorithm 1: How does one compute the function $\phi_\epsilon$ efficiently? Is there an expression for the TV distance between Mallows models that is being used here and also how is the supremum calculated? It would be good see some discussion regarding the running time of the algorithms.

Moreover, it might be mathematically more accurate to define $\phi_\epsilon$ as $\inf_{\pi_0 \in S_m} \sup_{\phi \in [0, 1]} \{ d_{TV}(\mathcal M_{1, \pi_0}, \mathcal M_{\phi, \pi_0}) > \epsilon \}$.


Page 4, Line 174: The notation $X_{i(i+1)}^{\ell}$ is confusing. It would be better to have something like  $X_{i, (i+1)}^{\ell}$


Page 5, Algorithm 2: Why does the algorithm only consider inversions between consecutive pairs $i$ and $ i+1$? Why not consider  inversions between all pairs $i, j$? Does considering consecutive pairs simplify the analysis of the algorithm?


Page 5, Theorem 5: I don't  understand how $\mu^2$ depends on $m$ and $\epsilon$. It should be explained why $\mu$ is constant in $m$. Moreover, what is the dependence of $\epsilon$ that comes out this sample complexity? Also, it is obvious that $\phi_\epsilon < 1$?


The discussion following Theorem 5 is unclear and needs to be clarified.


The classical problem where the central ranking is known should be highlighted more. For example, discussion in Section 7.2 is nice and very relevant, and should probably be mentioned earlier in the paper.






**Time Spent Reviewing:**

72 hours

---

> ### Author Response · Authors · 2021-08-09
> **Response to Reviewer r9MB**
>
> Thank you for your constructive feedback.
> * We are aware of these very relevant papers mentioned by the reviewer, and we will add them to the related work section of the final version where there is enough space to include them.
> * Computation of $\phi_{\epsilon}$: the total variation distance between two Mallows model with the same central ranking can be computed in quadratic time in $m$, since the permutations can be grouped by their Kendal distance from the central ranking and then the total variation distance boils down to computing $\sum_{i=0}^{m*(m-1)/2} \vert1-\phi^i \vert * M(i,m)$ where $M(i,m)$ is the $i$th Mahonian number of order $m$ and the normalization $Z(\phi)$. The Mahonian number can be computed based on recursion (Knuth 1962, Vol 3) and the normalization of the Mallows model can be computed in closed form (Fligner and Verducci 1986). Furthermore, the total variation distance for Mallows model from the uniform distribution is monotone in $\phi$, thus binary search can be applied to find $\phi_\epsilon$. We will add a remark on that. In our experiments, we assessed the power of algorithms for a given sample size, which is a more realistic scenario in practice. Thus the running time consists of computing the pairwise statistics which can be done in linear time in $m$ and in the sample size.
> * Regarding $\inf_{\pi_0 \in S_m } \sup_{\phi \in [0,1]} d_{TV} ( \mathcal M_{1, \pi_0}, \mathcal M_{\phi, \pi_0} )$: we would like to point out that the value of $d_{TV} ( \mathcal M_{1, \pi_0}, \mathcal M_{\phi, \pi_0} )$ does not depend on the choice of $\pi_0$.
> * Regarding consecutive pairs: the reason why only consecutive pairs are taken is because then the pairwise statistics are independent of each other, and as the reviewer pointed out, this simplifies the analysis.
> *$\phi$ is always in $[0, 1]$, with $\phi = 1$ corresponding to the uniform distribution. So, $\phi_\epsilon$ must be  less than 1, in order to get $d_{TV}  ( \mathcal M_{1, \pi_0}, \mathcal M_{\phi_{\epsilon}, \pi_0} ) >= \epsilon$ to the uniform distribution.
> * $\mu$ depends on both $\epsilon$ and $m$ in general. Unfortunately, no closed form of $\phi_\epsilon$ as a function of $\epsilon$ and $m$ is known (and does not seem manageable to obtain such a good approximation for all values of $\epsilon$), which would allow us to fully quantify the dependence. But, if $\epsilon = \Omega(1/\sqrt{m})$, then $\phi_\epsilon \leq 1 - \frac{c \epsilon}{m} \leq e^{-c \epsilon / m}$, for some constant $c > 0$. Then, $\mu \geq \frac{1-e^{-c \epsilon/8}}{1+e^{-c \epsilon/8}}$, which does not depend on $m$. Thus, unless $\epsilon$ is extremely small, i.e. unless $\epsilon = o(1/\sqrt{m})$, $\mu$ is indeed constant in $m$.
> * The discussion after Thm 5 attempts to highlight that (i) if $m$ is relatively small (so small that $m_1$ becomes negligible), a more careful analysis of UNIF may result in an improved sample complexity; and (ii) that the sample complexity of UNIF is upper bounded by a family of functions that have the same form as that in the last inequality of the proof of Thm. 5, in Appendix B, but use different values of $m_1$ and $\mu$ (in general, one can increase $m_1$ from $m/8$ up to $m$, by subsequently decreasing the exponent of $\phi_\epsilon$ in $\mu$ from $m/8$ to $1$. We will extend and clarify those points in the final version of our work.

---

### Official Review · Reviewer_C58N · 2021-08-02

**Rating:** 6
**Confidence:** 3

**Summary:**

The paper studies the problem of uniformity testing in ranking data: that is testing whether an unknown distribution is uniformly random over possible rankings OR it is epsilon away from it (in total variation distance). Noting that this problem in general is quite hard, they consider the special case where the alt-hypothesis is that the unknown distribution is a single parameter  Mallows model (i.e., exponential decay of permutations around an unknown fixed ranking).

The key contributions of the paper include a testing method for this problem with O(1/\sqrt(m)) samples. They also show that they can adapt their algorithm to enable local differential privacy.

**Limitations And Societal Impact:**

Yes

**Main Review:**

As I mentioned, the central contributions of the paper include a testing method for this problem with O(1/\sqrt(m)) samples (they also show only two samples are needed when m, the number of items,  is large enough -- which also follows from the above result (?). Why not just present this one result instead of two). They also show that they can adapt their algorithm to enable differential privacy.

The underlying idea behind this (non-private) result is quite simple to describe: partition the items into m/2 random disjoint pairs and construct a random variable that accounts for the deviations between the pairs of items. The idea is if the null hypothesis is satisfied value of the above random variable is large, else if the unknown distribution follows the mallows model, then the value is significantly small.

I like the simplicity of the algorithm -- which means it can useful in practice. On the flip side, I wonder if the assumption on alternative hypothesis class being a single variable exponential around a fixed ranking makes the problem too simple and uninteresting. The proof structure seems to following standard concentration analyses.

**Time Spent Reviewing:**

2-3 hours

---

> ### Author Response · Authors · 2021-08-09
> **Response to Reviewer C58N**
>
> Thank you for your positive feedback.
> * The reason why we present both uniformity testing algorithms is that the 2SAMP algorithm does not work for small $m$, but the UNIF algorithm does work for arbitrary $m$. In addition to this, the 2SAMP algorithm achieves better performance than UNIF in practice when we compared their performance in the experiments with large enough $m$, see Figure 1.
> * Regarding the assumption on the alternative hypothesis: we would like to emphasize that the alternative hypothesis class consists of all Mallows models with arbitrary central ranking. That is why the random selection of a ranking in Line 2 of Algorithm 2 is necessary to handle this large class of models that constitutes the alternative hypothesis in an efficient way. Moreover, our results can be applied with the Generalized Mallows model as alternative hypothesis class too, which family of models is a very natural extension of the Mallows model, however the constants in the sample complexity bounds will become more complex. We make sure that a discussion will be added to the final version and the analysis to the appendix.

---

### Decision · Program_Chairs · 2021-09-28

**Decision:**

Accept (Poster)

**Comment:**

Reviewers appreciated the natural and well-motivated problem under consideration, the simplicity and practicality of the proposed algorithms, and the general technical execution of the paper. They raised a few low-level technical concerns, primarily about notation and about details of subroutines used in the algorithms. These were sufficiently addressed in author feedback so as to reach a consensus to accept during discussion.

**Consistency Experiment:**

NeurIPS has a long history of experimentation. In 2014, NeurIPS ran an experiment in which 10% of submissions were reviewed by two independent committees to quantify the randomness in the review process. This year, we repeated a variant of this experiment to see how the quality of the review process has changed over time.  This paper was part of the experiment and was therefore assigned to two committees (consisting of reviewers, an Area Chair, and a Senior Area Chair) that reached independent decisions.  If both committees made the same recommendation, this recommendation was followed. If a single committee recommended acceptance, the paper was accepted (with the exception of a few cases in which the other committee identified what we considered a fatal flaw, e.g., an error in a key result).

Both committees reached the same decision: **Accept (Poster)**

The other committee assigned to the paper recommended **Accept (Poster)**.  You can find the other set of reviews, along with any follow up discussion with the authors here:
https://openreview.net/forum?id=lMrwT4C93eT